# Patient and Provider Perspectives on Emergency Department Care Experiences among People with Mental Health Concerns

**DOI:** 10.3390/healthcare10071297

**Published:** 2022-07-13

**Authors:** Carolina Navas, Laura Wells, Susan A. Bartels, Melanie Walker

**Affiliations:** 1Department of Public Health Sciences, Queen’s University, 62 Fifth Field Company Lane, Kingston, ON K7L 3N6, Canada; carolina.navas@queensu.ca (C.N.); susan.bartels@queensu.ca (S.A.B.); 2School of Medicine, Queen’s University, 15 Arch Street, Kingston, ON K7L 3N6, Canada; lwells@qmed.ca; 3Department of Emergency Medicine, Queen’s University, 76 Stuart Street, Kingston, ON K7L 2V7, Canada

**Keywords:** emergency department, mental health, care experiences, quality improvement, North America

## Abstract

Emergency departments (EDs) are an important source of care for people with mental health (MH) concerns. It can be challenging to treat MH in EDs, and there is little research capturing both patient and provider perspectives of these experiences. We sought to summarize the evidence on ED care experiences for people with MH concerns in North America, from both patient and provider perspectives. Medline and EMBASE were searched using PRISMA guidelines to identify primary studies. Two reviewers conducted a qualitative assessment of included papers and inductive thematic analysis to identify common emerging themes from patient and provider perspectives. Seventeen papers were included. Thematic analysis revealed barriers and facilitators to optimal ED care, which were organized into three themes each with sub-themes: (1) interpersonal factors, including communication, patient–staff interactions, and attitudes and behaviours; (2) environmental factors, including accommodations, wait times, and restraint use; and (3) system-level factors, including discharge planning, resources and policies, and knowledge and expertise. People with MH concerns and ED healthcare providers (HCPs) share converging perspectives on improving ED connections with community resources and diverging perspectives on the interplay between system-level and interpersonal factors. Examining both perspectives simultaneously can inform improvements in ED care for people with MH concerns.

## 1. Introduction

Mental health (MH) concerns represent one of the top causes of disability in North America, with approximately 1 in 5 adults experiencing a mental illness in any given year [1,2]. A 2018 study examining the prevalence of MH concerns estimated the weighted average 12-month prevalence of MH concerns in North America to be 22.5%, highlighting the significant impact MH concerns have on public health [3]. MH concerns affect behaviours, moods, and thought processes, and range in severity [1]. Mood disorders, anxiety disorders, post-traumatic stress disorder (PTSD), schizophrenia, and borderline personality disorder (BPD) are among the MH diagnoses most commonly reported in North America [1,2]. Suicidal ideation (SI) and substance use disorders often co-occur with MH concerns [1].

Individuals who identify as being a part of equity-deserving groups, including those who identify as ethnic minorities, those who are of low socioeconomic status, and those who are vulnerably housed, are disproportionately negatively impacted by MH [1,2,4,5]. The COVID-19 pandemic has further exacerbated the impact of mental illness in North America, with the proportion of Canadians reporting excellent or good MH decreasing from 68% in 2019 to 55% in June of 2020 [6] and a 93% increase in the number of Americans seeking help for anxiety and depression [7].

People with MH concerns receive most of their care at the community level through outpatient services. However, urgent care at the emergency department (ED) is an important aspect of the MH care continuum [8]. A wide range of MH concerns are addressed at the ED, with suicidal ideation and attempts being leading causes of ED help-seeking among MH patients [9,10,11]. EDs can provide access to psychiatric assessment as well as opportunities for connections and referrals to community organizations that are otherwise often difficult to obtain [8]. MH care in community settings can also be difficult to efficiently access due to long wait times for services [1], inadequate insurance coverage [4], and lack of public funding for services [4,5]. As a result, people with MH concerns visit the ED at disproportionately higher rates for both urgent and non-urgent concerns [12,13,14]. In fact, people with MH concerns are almost five times more likely to be frequent ED users (i.e., individuals with at least four ED visits in the past year) in comparison to patients with no MH concerns [15]. National Canadian and American data shows that MH concerns represent 9.8% and 12% of all ED visits, respectively [16,17]. This high frequency of use contributes significantly to overcrowding and congestion in the ED [18,19].

Treating MH concerns in the ED can be challenging given the volume of total patients, limited face-to-face time between patient and provider and complexity of psychiatric cases [19,20]. Additionally, perceived negative attitudes and stigmatizing behaviours among health care providers (HCPs) can interfere with effective treatment for people with MH concerns [19,21,22,23]. These challenges are further compounded by system-level issues that prevent HCPs from providing the quality of care that is necessary for people with MH concerns, including lack of appropriate training on MH, lack of standardized approaches to risk assessment and treatment of MH concerns, and reduced access to MH specialists in the ED [24,25]. Despite these barriers, people with MH concerns continue to seek support through EDs because, for most individuals in a MH crisis, the ED is the most accessible way to receive immediate care.

Research demonstrates that patient experience and satisfaction are important outcomes of interest among this population [24]. If people with MH concerns, who already experience higher rates of morbidity and mortality, are satisfied with their healthcare, they will be more likely to seek help when they feel unwell and are less likely to require hospital readmission [24]. However, patient ED care experiences are underrepresented in the current literature [24]. Although there have been individual studies and reviews on either patient or provider perspectives of ED care for people with MH concerns, the experiences of both have not been reviewed collectively. It is important to systematically evaluate the totality of evidence including perspectives of patients and HCPs to better understand the perceived facilitators and barriers to good ED care experiences and to identify strategies that will improve care for people with MH concerns. This literature synthesis, therefore, aims to answer the following questions: what are the ED care experiences, from both the patient and provider perspective, of people with MH concerns in North America, and what are common barriers and facilitators to optimal care for people with MH concerns? It is hypothesized that ED care experiences will be mostly negative from both perspectives, with patients reporting largely on stigmatization and judgement, and providers reporting largely on difficulty providing care and organizational restraints.

## 2. Materials and Methods

### 2.1. Search Strategy

We searched Medline and EMBASE for studies that involved patient and/or provider perspectives on MH care in the ED using key words such as “psychiatric patient”, “mental disorders”, “emergency department”, “emergency room”, “patient satisfaction”, and “professional-patient relations”. Key words were identified in collaboration with a professional Librarian. The search was not limited by study design, type of ED, gender, age, or MH concern. The reference lists of relevant articles identified in the search were manually examined for additional articles not identified through the original search to minimize publication bias.

### 2.2. Inclusion and Exclusion Criteria

We included research that was: (a) published between 1 January 2005 and 5 June 2021, (b) published in English, (c) based in Canada or the United States, (d) primary studies, (e) based in the ED and (f) focused on patient and/or provider perspectives. Article titles and abstracts were screened in duplicate according to pre-specified inclusion and exclusion criteria (C.N. and L.W.). A full-text screening of all articles that met inclusion criteria was subsequently conducted by two reviewers (C.N. and L.W.).

### 2.3. Assessment of Methodological Quality

Relevant study information, methodology, and results were extracted independently by two researchers (C.N. and L.W.), and a quality assessment of included studies was performed. The extracted data were inputted into a spreadsheet, organized by categories that were modelled off of review articles of similar style in the same field of research [21]. A quality assessment of each study was conducted as data were extracted using an adapted CASP checklist created by the researchers (C.N. and L.W.) containing categories developed from recommendations and guidelines for critically appraising qualitative literature [26,27]. These categories included: author of article, perspective type (patient, provider, or both), country where research was conducted, study objective, participants, study design, study methods, sample characteristics, main findings, and main sources of bias and limitations.

### 2.4. Thematic Analysis

Data were extracted from articles as described in Section 2.3 and analyzed using an inductive thematic synthesis of the study results [28]. Two researchers (C.N. and L.W.) conducted iterative reviews of the extracted data to inductively identify common barriers and facilitators that influence ED care among people with MH concerns, with adjudication by researcher consensus (C.N. and L.W.) in areas of disagreement. These themes and sub-themes were agreed by the researcher team (C.N., L.W., S.A.B., and M.W.) prior to being amalgamated into the themes and sub-themes identified in this paper.

## 3. Results

The search identified eight-hundred and twenty-three articles, of which three hundred were duplicates, leaving five-hundred and twenty-three articles eligible for screening. Sixty-one articles met all inclusion and exclusion criteria after abstract screening, and sixteen articles met all criteria after full-text screening. An additional article was identified through manual searching of reference lists for a final total of seventeen articles included in the review.

### 3.1. Quality Assessment

Seventeen articles were included in this review following title, abstract, and full-text screening, as well manual searching of relevant reference lists (Figure 1). A full description of study characteristics and critical appraisal for included articles is found in Table 1. Ten articles were from the United States [29,30,31,32,33,34,35,36,37,38] and seven from Canada [39,40,41,42,43,44,45]. Ten papers explored patient perspectives [29,31,33,34,37,38,41,42,44,45], six of which focused on patients with any MH concern (with or without concomitant substance use disorder) [30,32,34,35,36,37], two focused on patients who were physically restrained [33,38], one focused on patients with a suicide attempt [31], and one focused on patients with anxiety and depression only [29]. Six papers explored provider perspectives, including resident and attending physicians, nurses, nurse practitioners, social workers, rehab instructors, nurse aides, managers, and addiction team members, the most common of which were the perspectives of nurses and attending staff physicians [30,32,35,36,40,43]. One study explored both patient and provider perspectives [39]. All studies were based exclusively in the ED, with the exception of one that explored and compared provider perspectives across multiple settings, in addition to the ED [40].

Eleven studies employed descriptive study designs [30,34,36,37,38,39,41,42,43,44,45]; eight studies used qualitative designs, including phenomenological [34,36,37,45], interpretive description [41], grounded theory [38], and case studies [39,43], which employed a variety of qualitative data collection methods, such as semi-structured interviews [38,41,43,45], focus groups [36,37,39], and secondary analysis of qualitative interview data [34]; and three studies used descriptive cross-sectional designs with no comparison group, using either mixed methods [42,44] or quantitative [30] approaches. Six articles used analytic study designs [29,31,32,33,35,40]; three studies used analytic cross-sectional designs with comparison groups [29,31,40], employing both mixed methods [31] and quantitative methods [29,40] for data collection; two studies used a prospective cohort design with quantitative methods of data collection [32,33]; and one study used a quasi-experimental design within a convergent mixed methods approach to data collection [35].

### 3.2. Thematic Analysis

A thematic analysis of both patient and provider perspectives across studies revealed three key themes, each with their own sub-themes—interpersonal factors, environmental factors, and system-level factors—visually depicted in Figure 2 as an adaptation to Bronfenbrenner’s social–ecological model [46].

### 3.3. Interpersonal Factors

Interpersonal factors, such as the thoughts, actions, and behaviours between individuals that affect optimal ED experiences, emerged as a major theme in the literature. Three key sub-themes related to interpersonal factors were identified: communication about care, patient–staff interactions, and attitudes and behaviours about care. For this analysis, attitudes and behaviours included identified feelings or thoughts held by patients and/or providers about an object, process, or person, and how these attitudes were reflected in behaviours or acts by patients and providers [47].

#### 3.3.1. Communication about Care

##### Patient Perspectives

Clear and timely communication about treatment was highly valued by patients [37]. People with MH concerns described their interest in understanding the ED care process, including details about the specific HCPs that they see and the various treatment options available [37]. Thomas et al. (2018) noted that people with MH concerns are often familiar with their diagnoses, especially if they are frequent users of the ED, and appreciated explanations for the course of treatment chosen [37]. However, many participants reported dissatisfaction with the level of communication from ED HCPs [31,34,37,38,41].

Patients who have been restrained and involuntarily committed to the hospital during an ED visit described a lack of understanding and poor communication from HCPs regarding the decision to use restraints [37,38]. These patients reported a desire to receive more information from staff about what led to the decision to use restraints [37].

Use of appropriate terminology, accessible to patients and their supporters, was also identified as being important. Some people with MH concerns reported that HCPs used medical jargon, which led to worsened mental states due to feelings of stress and loss of control [31,34].

##### Provider Perspectives

Staff described significant difficulty communicating with frequent users of the ED, especially those with co-occurring MH and substance use disorders, BPD or suicidal behaviours and ideation [43]. ED HCPs indicated that they had less confidence with these cases and had trouble addressing the primary concern when patients presented with multiple associated pathologies and symptoms [43]. Providers also described not knowing whether they were being effective or not when speaking with people with MH concerns, which further hindered productive communication [36].

#### 3.3.2. Patient–Staff Interactions

##### Patient Perspectives

Positive experiences with ED staff were highly valued by people with MH concerns, including instances of feeling cared for and respected, being treated well and fairly, listened to and advocated for by the HCPs, and having familiarity with staff members who understood their needs [39,41,42,44]. However, the vast majority of the literature reported negative experiences among people with MH concerns with respect to interactions with staff members [31,33,34,38,41,45]. In particular, patients frequently described feeling judged and disrespected by HCPs due to their MH concern [31,34,41]. Similar experiences were identified by patients who visited the ED following a suicide attempt, with over half of patients feeling punished and stigmatized by staff due to their suicide attempt [31]. Lack of consideration for needs and unsympathetic care from HCPs also encompassed negative experiences of people with MH concerns [29,37,41,45]. People with MH concerns at the ED described HCPs as “pretty cold” and “nasty” and reported feeling as though staff did not care about them [34,45]. In one study, an individual diagnosed with PTSD described feeling distressed and triggered when asked to remove her clothing in order to put on a hospital gown; this was perceived as a lack of caring from the patient’s perspective [24]. Additionally, multiple studies discussed a lack of culturally appropriate care which led to overall negative patient experiences [31,41] such as one patient with a MH concern being met with contempt due to his cultural practice of fasting. [41].

##### Provider Perspectives

The perception of HCPs interactions with people with MH concerns in the ED were also largely negative [35,36]. Isbell et al. (2019) reported that providers described general negativity in encounters with people with MH concerns, with only a small proportion feeling positively about their interactions with this patient population [35]. The emotions reported by providers during encounters with people with MH concerns included anger, sadness, helplessness, empathy, fear, and anxiety [35]. Self-reported anger was associated with lower perceived engagement in encounters that HCPs had with people with MH concerns (*p* < 0.01) [35]. Other difficulties interacting with people with MH concerns described by Plant and White (2013) included not receiving acknowledgement or feedback when interacting with this population, not being able to see the end result like they do with other presentations or patients, and seeing people with MH concerns as “not an easy fix” [36].

#### 3.3.3. Attitudes and Behaviours

##### Patient Perspectives

People with MH concerns frequently described being treated differently and experiencing stigmatization because of their MH concern [29,32,34,39,45]. Harris et al. (2016) described that some people with MH concerns reported feeling judged by staff, specifically physicians, who made negative assessments about them based only on their MH concern [34]. People with MH concerns described that once they were deemed to be a patient with a MH concern, their presenting concerns were overlooked and triaged lower, even if they were presenting with a physical health condition—a concept known as diagnostic overshadowing [39]. Abar et al. reported strong correlations between having anxiety and, independently, depression and feelings of non-responsiveness from HCPs (anxiety: r = 0.27 (*p* < 0.001); depression: r = 0.33 (*p* < 0.001)) as well as feelings of embarrassment about potential illnesses (anxiety: r = 0.28 (*p* < 0.001); depression: r = 0.22 (*p* < 0.001)) [29]. Despite these perceptions, people with MH concerns viewed their visits to the ED as unavoidable and would find alternative strategies to avoid being labelled as a MH patient in the ED, such as not disclosing their psychiatric history or bringing someone to advocate on their behalf [39,44].

##### Provider Perspectives

Plant and White (2013) found that providers experienced challenges triaging people with MH concerns, felt more comfortable treating and managing physical health presentations and feared that they overlooked illnesses due to perceived manipulative behaviours among people with MH concerns [36]. Providers also expressed concerns that people with MH concerns were ‘abusing the system’ and taking time and resources away from other patients, such as those who had experienced physical trauma [36]. In the study by Isbell et al. (2019), HCPs perceived overall negative experiences with people with MH concerns, citing patient behaviours as demanding, manipulative, and entitled in 52% of encounters [35]. Overall, only 23% of providers perceived encounters with people with MH concerns to be positive [35].

### 3.4. Environmental Factors

Environmental factors, including an individual’s immediate surroundings, emerged as a key theme. Three sub-themes that related to the influence of the environment on ED care included accommodation and basic needs, wait times and time spent receiving care, and involuntary admission and restraint use.

#### 3.4.1. Accommodations and Basic Needs

##### Patient Perspectives

Some of the most extensive findings include negative experiences for people with MH concerns associated with the hospital environment and accommodations [33,34,37,39,44,45]. Patients reported that the clinical, fast-paced nature of the ED, including smells and sounds, made them feel frightened and overstimulated [31,39]. People with MH concerns reported that the demeanour of staff members often overwhelmed them, leading to increased feelings of anxiety, panic, and agitation [31,39]. This sensory overload in the ED was triggering for some patients who had previously traumatizing experiences, and these sights and sounds had the potential to unearth negative memories [34]. Feelings of isolation, loneliness, worry, and confusion were reported by people with MH concerns when seeking care at the ED [34,38,39]. Some patients described increased levels of anxiety, feelings of abandonment, a lack of ability to self-advocate, as well as the exacerbation of MH symptoms when left alone to wait for psychiatric staff after being triaged by ED staff [34,39].

People with MH concerns also reported that having access to basic necessities provided them with a sense of dignity and respect; however, most patients described experiences of their needs being left unmet, including poor access to meals, hygienic spaces (i.e., bathrooms), and clean gowns [34,39,44]. Patients also reported feeling uncomfortable, overwhelmed, and stressed when not given privacy during triage or when being placed in a shared room with another patient, which can further exacerbate their distress [34,37,39].

##### Provider Perspectives

Providers also recognized the importance of privacy and accommodation needs for people with MH concerns. In their study exploring barriers and facilitators to ED care for people with MH concerns, Fleury et al. (2019) identified that the existence of management protocols and designated service corridors for people with MH concerns with physical conditions facilitated ED care [43]. However, the HCPs also felt that a lack of appropriate physical space to interview people with MH concerns was a barrier to providing effective care [43]. Plant and White (2013) also identified frequent barriers to finding appropriate placement and privacy for people with MH concerns in the ED [36].

#### 3.4.2. Wait Times and Time Spent Receiving Care

##### Patient Perspectives

Many people with MH concerns reported long delays in being connected with HCPs after triage and general unavailability of specialized psychiatric professionals [34,39,44,45]. In one study, a typical wait time of 8–10 h was reported by people with MH concerns [39]. In these situations, people with MH concerns reported feeling unworthy of care or forgotten about, and often believed that MH concerns were ranked last during triaging, further contributing to their negative perception of the ED [39]. Some people with MH concerns also reported being asked to wait for psychiatric staff following triage, just to be triaged again for their initial medical concern, further prolonging their overall wait times [39]. Some patients reported that they “gave up and went home after long waits”, therefore not receiving the care they needed [39]. Long wait times also created secondary issues for people with MH concerns, as some individuals reported that being left alone to wait in the ED exacerbated their anxiety and distress, thus contributing to a worsened mental state (26). As a result of these experiences, people with MH concerns described a need for improved triage procedures in the ED for their medical needs to be met in a timely manner [34].

##### Provider Perspectives

Providers also reported several barriers related to wait times and time spent delivering care for people with MH concerns [32,36]. According to ED staff, any issues with bed or staff availability further increased overall wait times for these patients by 92.5 min (SE = 34.9, *p* < 0.01) and 80.5 min (SE = 41.3, *p* = 0.05), respectively, often leading to dissatisfaction [32]. Providers also described feelings of pressure to evaluate and process patients quickly, as well as significant organizational restrictions on length of stay, which make it difficult to provide adequate time and services to people with MH concerns [43].

#### 3.4.3. Involuntary Admission and Restraint Use

##### Patient Perspectives

People with MH concerns reported negative experiences with involuntary admission to the ED and use of chemical or physical restraints [34,37,38]. Patients who had been restrained described feeling as though they were criminals, not patients, and generally felt a loss of freedom and dignity [34,37,38]. Upon being placed in restraints, people with MH concerns reported feeling confused, frustrated, and isolated [38]. Wong and colleagues (2020) found that experiences with restraints, whether physical or chemical, were traumatic and had long-lasting negative impacts on the patient’s trust, confidence, and engagement in the healthcare system [38]. Additionally, these experiences often made patients worry about what they do or say in the presence of HCPs, further contributing to mistrust between the patient, HCPs, and the healthcare system [34]. In a prospective cohort study comparing the experience of people with MH concerns who were restrained to people with MH concerns who were not restrained, the use of physical restraints was found to be significantly associated with less frequent attendance at outpatient psychiatric appointments (OR = 0.38, 95% CI 0.16–0.89) [33]. Despite these negative experiences, a select number of patients indicated an understanding of the intention of HCPs and agreed that the use of restraints was justified in their case [33,38].

##### Provider Perspectives

One study evaluated provider perspectives on restraint and seclusion use across hospital units, including EDs, by comparing the perception of staff members who were high users of restraints to low users [40]. Use was highest in EDs, and staff members were more likely to restrain a patient if other staff members were expressing anger or aggression [40]. Additionally, the behaviour of the patient also impacted restraint use, with staff members being more likely to use restraints if the patient was demonstrating aggression towards themselves [40].

### 3.5. System-Level Factors

System-level factors, including institutional and hospital level practices and policies, was another key theme that emerged. Three sub-themes were identified: discharge planning and coordination of care, resource and policy constraints, and knowledge and expertise.

#### 3.5.1. Discharge Planning and Coordination of Care

##### Patient Perspectives

People with MH concerns noted that the ED can act as a connection to other HCPs in the community, which can prevent future reliance on the ED [41,45]. Prior to discharge from the ED, people with MH concerns greatly appreciated being told about referrals and potential organizations for follow-up care in the community, and also preferred having the paperwork done for them to facilitate the process [37,41]. However, patients generally reported that there was inadequate information provided at discharge about programs and services outside of the hospital [41,43]. People with MH concerns reported feeling as though they were “kicked out” of the ED without their concerns being adequately addressed, therefore creating a cycle of ED help-seeking behaviours and mental states that worsen with early discharge [41,45]. Despite these experiences, people with MH concerns continued to seek support through EDs because it was the only place to receive care, particularly after hours [39].

##### Provider Perspectives

The use of multidisciplinary teams and interprofessional providers were identified by HCPs as facilitators to ED care for people with MH concerns [43]. ED providers who work with people with MH concerns indicated that liaison nurses, social workers, assertive community treatment teams, and intensive care management facilitated and ensured adequate follow up for people with MH concerns [36,43]. However, providers also reported that there was often limited knowledge of community organizations or poor connections in-hospital with these community organizations, which acted as a barrier to providing adequate and meaningful referrals [36,43]. Additionally, referral processes were often further delayed due to overcrowding across MH services such as addictions programs, primary care, private professionals, and housing services [43]. Providers expressed a desire for consistent teamwork and coordination in order to address these barriers [36].

#### 3.5.2. Resource and Policy Constraints

##### Patient Perspectives

Rules and regulations, such as the inspection of personal belongings, surveillance, and smoking restrictions, were all found to contribute to negative ED care experiences for people with MH concerns [42]. In an American cross-sectional study assessing perceived barriers to ED care among people with anxiety and depression compared to controls, both controls and people with MH concerns described barriers to care related to resources and policies, such as healthcare bills, scheduling appointments, and transportation [29]. However, patients with both anxiety and depression had higher mean levels of self-reported barriers to care (M = 1.12, SD = 0.66) compared to people with anxiety alone ((M = 0.63, SD = 0.31) or depression alone (M = 0.55, SD = 0.59), or compared to controls (M = 0.32, SD = 0.39), *p* < 0.001 [29]. In this study, people with MH concerns reported more barriers to finding transportation and confusion around appointment scheduling [29].

##### Provider Perspectives

Fleury et al. (2019) noted that having best practice guidelines for triaging of people with MH concerns was an effective organizational facilitator for ED care [43]. Issues with the ED system, the hospital, and healthcare in general were main themes described by HCPs for all patient encounters, but especially for negative encounters with people with MH concerns [35]. Adhering to policies and regulations, poor clinical management, and limited staff recruitment have all been identified by ED staff as organizational barriers to providing care [36,43]. ED managers described difficulty in recruiting competent staff who had experience in MH care, including physicians, nurses, and other HCPs [36,43]. Additionally, quick staff turnover and lack of staff availability often created periods of shortages of HCPs in the ED [43]. For example, one hospital reported that they did not have psychiatrists or liaison workers available during some evenings and weekends, even though they had been identified as crucial care team members [43].

#### 3.5.3. Knowledge and Expertise

##### Patient Perspectives

Interacting with ED staff who had knowledge of MH contributed to positive experiences for people with MH concerns [42]. Conversely, a lack of MH knowledge and expertise, specifically around PTSD, BPD, and MH in young adults, disorders co-occurring with physical/developmental disabilities, and substance use, all contributed to negative patient experiences [39]. In Clarke et al.’s (2007) study, people with MH concerns expressed a desire for physicians and non-ED staff (e.g., police, paramedics) to have improved training on these topics, and felt this would improve their future ED experiences [39].

##### Provider Perspectives

Lack of training, knowledge, and expertise in MH were identified as significant barriers for ED staff, leading them to feel as though they did not have the necessary skills to properly address patient issues [30,44]. Betz et al. (2013) surveyed the attitudes of ED HCPs, and their findings demonstrated knowledge gaps in screening, assessment, and support for suicidality, with more nurses (*n* = 112, 37%) than attending (*n* = 11, 8.1%) or resident physicians (*n* = 12, 7%) reporting screening most or all patients for suicidal ideation [30]. ED nurses and physicians reported that they did not feel confident assessing risk level among people with MH concerns and felt unable to provide counselling or create safety plans [30]. Similar perspectives were shared by Plant and White, where ED nurses described continuous challenges with identifying, triaging, treating, and caring for people with MH concerns, as well as a lack of understanding of and education on the long-term consequences of MH concerns [36]. Staff also reported that poor administrative support prevented implementation of interventions that would improve suicide care in the ED. In fact, Betz et al. (2013) showed that less than a quarter of HCPs believed that support for suicidal patients was a top priority in their hospital [30].

## 4. Discussion

This review demonstrates that a majority of ED experiences are perceived as negative by people with MH concerns. These findings align with recent literature describing predominantly negative experiences of people accessing the ED for mental health crises [48,49]. Similarly, this review highlights challenging provider experiences when caring for people with MH concerns in the ED, which is reflective of recent literature summarizing the perspectives of providers of MH care in the ED [21]. To our knowledge, this is the first review to synthesize the existing evidence on both patient and provider perspectives of ED care experiences for people with MH concerns in North America. Patients described barriers and facilitators largely at the interpersonal and environmental levels, while providers reported a greater number of barriers and facilitators at the system level. Simultaneously examining patient and provider experiences helps to elucidate convergent and divergent perspectives, both of which are important to consider when developing appropriate intervention strategies to improve care for people with MH concerns.

### 4.1. Convergent Perspectives

Both patients and providers viewed the ED as a resource to connect people with MH concerns with the appropriate community care. However, they also described a significant lack of accessible community-based MH services, and a need for improved collaboration with EDs. Lack of information about resources, poor connections with hospitals, and overcrowding in community services were all reported by patients and providers as barriers to accessing appropriate services, all of which led to increased care-seeking by people with MH concerns in the ED. To improve ED care for people with MH concerns, investments should be made to increase the capacity of community services to support this population, in addition to strengthening and expanding existing connections between hospitals and community services [50,51]. This may facilitate a greater number of people with MH concerns seeking specialized community-based care and reduce non-emergent MH-related visits to the ED, thus decreasing ED wait times [50].

### 4.2. Divergent Perspectives

Patients perceived interpersonal factors, such as negative attitudes and behaviours from staff, as heavily influencing their care experiences. Simultaneous consideration of both patient and provider perspectives, however, suggests greater issues exist at the environmental and system levels. Insufficient specialized psychiatric staff, inadequate staff training and knowledge, and lack of organizational support often led to poor understanding of MH concerns and presentations, thus reinforcing inaccurate or negative stereotypes of people with MH concerns. These environmental and system-level issues manifest at the interpersonal level and are perceived by patients as poor communication, lack of caring from HCPs, diagnostic overshadowing, and ED environments that exacerbate their MH concern(s).

### 4.3. Approaches to Care

It has been well-established that diagnostic overshadowing can result in the misdiagnosis of people with MH concerns in the ED, causing delays in treatment and, resultantly, potential negative health outcomes [52,53,54]. Ensuring accurate diagnoses are made in the ED is therefore imperative to help improve care experiences for people with MH concerns [54]. Schefer and colleagues (2014) described the importance of psychiatric liaison staff in reducing the occurrence of diagnostic overshadowing among people with MH concerns by encouraging further assessments of patients that have been cleared by ED staff [53]. Educational interventions can also play an important role in addressing this issue. Clarke et al. (2006) showed that MH education for staff led to a significant decrease in the number of people with MH concerns that were under-triaged, and providers were more likely to assign the correct level of urgency to people with MH concerns [55]. Medical education programs that incorporate MH concerns and specialized training can, therefore, help ensure the proper and compassionate treatment of people with MH concerns [52].

Another strategy to improve care experiences for people with MH concerns is adopting trauma-informed care. Trauma-informed care is important to providing effective MH care as it minimizes the effects of traumatic experiences for patients [56,57]. Trauma-informed care can improve HCPs’ knowledge of MH, foster productive communication and relationships between providers and patients, and can reduce the use of harmful practices such as physical restraint [56,57]. Hall et al. (2016) demonstrated that adopting a trauma-informed care approach among nursing staff led to increased knowledge about the impacts of trauma and a broader understanding of the experiences of people with MH concerns [56]. However, the authors noted that prior to the widespread implementation of trauma-informed care interventions, there should be an appropriate attitude shift among staff and support at the organizational level [56].

Based on the perspectives of both patients and providers, it is evident that when people with MH concerns seek care in the ED, there are a series of complex, interconnecting factors that contribute to the care experience. These factors translate to a series of interpersonal, organizational, and systemic issues that create sub-optimal care experiences. Ongoing assessment, evaluation, and intervention are necessary across these levels, with a particular focus on the healthcare system as a whole, to address these long-standing barriers.

### 4.4. Strengths and Limitations

The qualitative studies included in this review have inherent limitations including lack of objective measures of association, small sample sizes, and the potential for self-reporting and interview biases. Additionally, the lack of a true control group consisting of people without MH concerns prevents the identification of unique experiences and barriers to ED care for people with MH concerns. The perspectives of individuals who identify as members of equity-deserving groups, including individuals who are ethnic or racial minorities, those who are of low socioeconomic status, and members of the 2SLGBTQIA+ community, were significantly under-represented in the studies included in this review. Most studies did not present their results stratified by MH diagnosis, which makes it difficult to accurately capture differences in ED care experiences across the MH spectrum, and therefore, difficult to provide recommendations for specific diagnoses. Lastly, the findings presented in this review may not be generalizable to all EDs, geographic areas outside of North America, or other populations that were underrepresented in the studies.

This systematic review employed rigorous a priori methodology using multiple databases to identify articles, inclusion/exclusion criteria to minimize selection bias, and a rigorous quality assessment informed by established guidelines and principles for reviewing qualitative literature. The thematic qualitative analysis was performed in duplicate to ensure high inter-rater reliability, and the inductive approach allowed the elucidation of themes not previously identified in the literature for people with MH concerns and HCPs who treat people with MH concerns in the ED. These results may help to inform quality improvement interventions to improve the care of a historically disadvantaged group from both patient and provider perspectives.

### 4.5. Future Directions

Future research on ED care for people with MH concerns should aim to capture experiences from a range of HCPs and patients, including people with MH concerns who also identify as members of equity-deserving groups, with the goal of making ED care more accessible for people with MH concerns [56]. Intersectionality, a concept originally described by Kimberle Crenshaw, identifies that the experiences of belonging to multiple equity-deserving groups cannot be considered in isolation, but rather as a unique experience [56,58,59]. Given the variations in experiences and health outcomes among people with MH concerns, an intersectional, equity-based approach focusing on the interconnectedness of race, ethnicity, socioeconomic status, and/or other social demographics is crucial to obtaining a comprehensive understanding of the experiences of people with MH concerns [59,60]. This should be performed both as a whole and for individual MH diagnoses to better understand the unique needs of people with MH concerns, and to make specific recommendations. Mixed-methods approaches should be used to better address the methodological shortcomings of solely qualitative or quantitative research by obtaining objective measures of experiences among larger populations while still capturing the nuanced subjective experiences of participants. Lastly, analytical studies that elucidate perspectives on ED care experiences among both patients with MH concerns compared with those who do not identify as having an MH concern will be useful to identify the barriers and facilitators unique to the care experiences of this important group of patients.

## 5. Conclusions

EDs act as an important part of the MH care continuum and provide immediate medical or psychiatric care. Both providers and patients alike identified similar facilitators and barriers related to the ED environment and access to community organizations. Patient and provider opinions diverged as they related to the interplay of interpersonal and system-level factors. These findings can help inform quality improvement initiatives to improve ED experiences for people with MH concerns and their HCPs. Future mixed-methods studies with control groups who do not identify as having a mental health concern can reduce the limitations inherent in the existing body of qualitative evidence and would contribute to this evidence base aimed at improving ED care for people with MH concerns.

## Figures and Tables

**Figure 1 healthcare-10-01297-f001:**
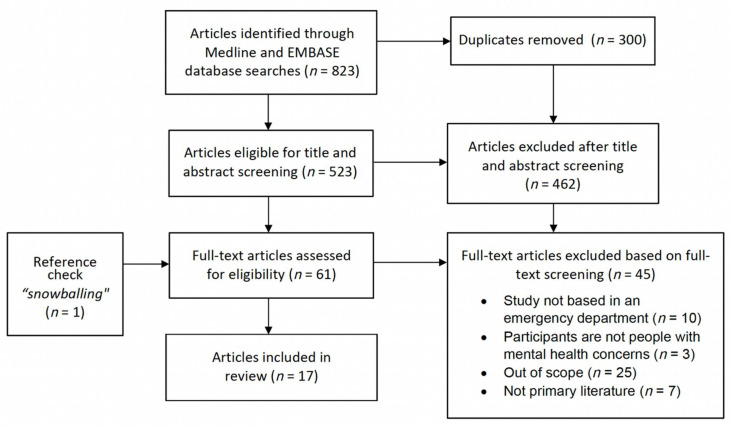
PRISMA diagram of the study review process.

**Figure 2 healthcare-10-01297-f002:**
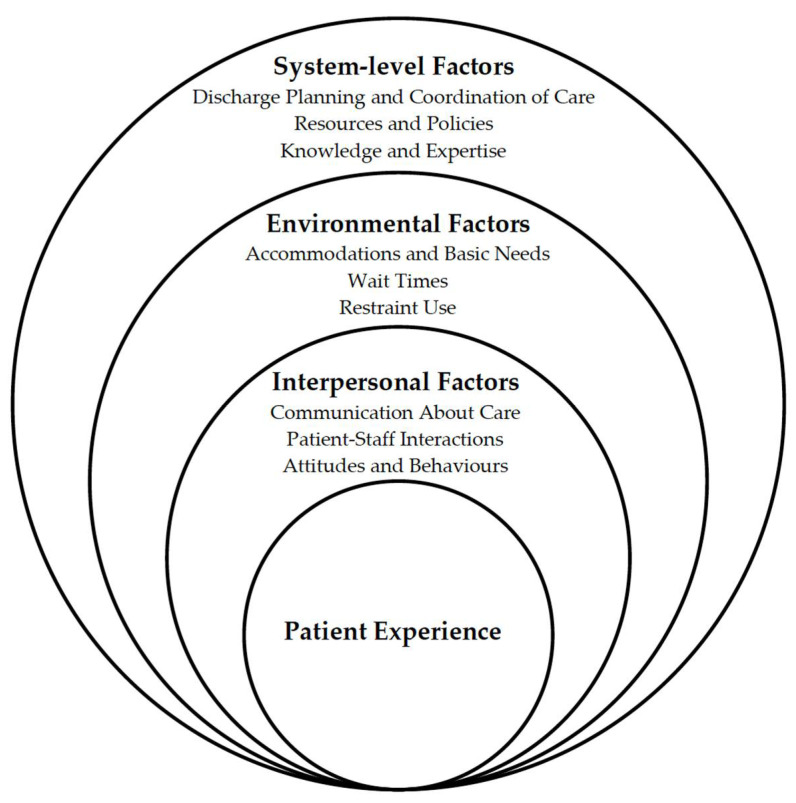
Conceptualization of the key themes identified through thematic analysis of patient and provider perspectives, adapted from Bronfenbrenner’s socio-ecological model [46].

**Table 1 healthcare-10-01297-t001:** Critical appraisal and quality assessment of articles included in paper (*n* = 17).

Author (Date)	Country	Objective	Participants	Study Design	Methods of Data Collection	Sample Characteristics	Main Findings	Limitations
**Patient Perspectives**								
Abar et al. (2017) [29]	USA	To relate anxiety and depression with ED utilization and perceived barriers to care	Patients aged 45–85 who presented to the ED	Analytic cross-sectional study comparing ED utilization and barriers among patients with anxiety, depression and both with patients without anxiety or depression	Questionnaire (web-based) collecting quantitative data on demographics, depression and anxiety screening and utilization and barriers to care	*n* = 251	Greater anxiety and depression scores were associated with more perceived barriers to care including healthcare bills, fear of serious illness and difficulty finding transportation	(1) Subject to self-reporting and recall bias; (2) older adults, children and youth, females, members of ethnic minority groups, and other MH diagnoses underrepresented
Cerel et al. (2006) [31]	USA	To explore the experiences of psychiatric patients and family members in the ED following a suicide attempt	Patients who presented to the ED for a suicide attempt compared to family and friends of consumers	Analytic cross-sectional study comparing the experiences of patients and family members/friends	Mixed-methods web-based survey with both yes/no (quantitative) and open-ended (qualitative) questions	Patients = 465 Family/friends of patients = 254	Most patients (*n* = 490) reported negative experiences during their time in the ED, involving perceptions of unprofessional staff behaviour, feeling lonely or ignored, feeling their suicide attempt was not taken seriously, and long wait times	(1) Subject to selection and recall bias; (2) males, older adults, children and youth, members of ethnic minority groups, single individuals underrepresented; (3) differences in experience by MH diagnosis were not collected
Currier et al. (2011) [33]	USA	To determine the viewpoints on quality of ED MH care, recollection of restraint episodes, and willingness to participate in outpatient psychiatric care for patients who are restrained	Patients who presented to the ED for MH care who were physically restrained compared with those who were not	Analytic prospective, two-arm cohort study	Quantitative data collected via structured, in-person, rather-administered questionnaire, followed by assessment of whether patients attend their outpatient appointment	*n* = 151 patients; Cases (Physically restrained patients) = 67; Controls (Not physically restrained patients) = 84	Both minority race and use of physical restraints were related to less frequent attendance at outpatient psychiatric appointment.	(1) Non-random allocation of groups and notable differences with regard to previous restraint use and MH diagnosis; (2) females, children, youth, and older adults underrepresented
Harris et al. (2016) [34]	USA	To describe the perceptions of ED visits by patients experiencing emotional distress, identifying themes that may guide nursing interventions that minimize stress and optimize treatment outcomes	Patients experiencing emotional distress in the ED	Descriptive qualitative—phenomenological	Secondary analysis of qualitative interview data	*n* = 9	Three major themes emerged: (1) emergency rooms are cold and clinical; (2) they talk to you like you’re a crazy person; (3) you get put away against your will	(1) Subject to recall bias and selection bias; (2) males, individuals without education and older adults underrepresented; (3) small sample size; (4) lack of objective measures of experiences; (5) no comparison group
Thomas et al. (2018) [37]	USA	To develop a better understanding of what patients with MH and substance-related disorders value to inform policy on psychiatric crisis services	Patients who had received psychiatric services in EDs or a community MH centre	Descriptive Qualitative—Phenomenological	Qualitative focus groups	*n* = 27	Themes that emerged included appreciation for feeling respected, basic comforts and shared decision-making as foundations of quality care; patients preferred community mental health centres over EDs for treatment	(1) Small sample size; (2) lack of objective measures of experiences; (3) no comparison group; (4) potential selection bias and recall bias; (5) differences in experience by MH diagnoses were not collected; (6) older adults underrepresented
Wong et al. (2020) [38]	USA	To characterize how individuals experience episodes of physical restraint during their ED visits	ED patients who had been restrained during a visit	Descriptive Qualitative—Grounded theory	Qualitative semi-structured interviews and survey with sociodemographic questions	*n* = 25 (57% non-responsive and 5% declining to participate) 16% had a mental illness only; 24% had an alcohol or drug use disorder only; 60% a combination	Most patients felt coerced to present to the ED, did not present willingly and reported negative outcomes related to their restraint experiences. Despite this, most patients expressed a desire for dignity, respect and compassion.	(1) Subject to selection bias; (2) small sample size; (3) lack of objective measures; (4) no comparison group; (5) females, members of ethnic minority groups, children, youth, and older adults underrepresented; (6) differences in experience by MH diagnosis were not collected
Digel Vandyk et al. (2018) [41]	Canada	To explore the experiences of patients who frequently present to the ED for mental health-related reasons	Patients who frequently present to the ED for MH-related reasons	Descriptive qualitative study—Interpretive Description	Semi-structured interview with sociodemographic survey	*n* = 10 6 had a primary diagnosis of borderline personality disorder, 2 of schizophrenia, 1 of bipolar diagnosis, 1 of substance use disorder, 6 had co-occurring substance use disorder	Key themes identified in the analysis included: (1) The Experience; (2) The Providers; (3) Protective factors.	(1) Small sample size; (2) lack of objective measures of experience; (3) no comparison group; (4) subject to social desirability bias and recall bias; (5) differences in experience by MH diagnoses were not collected; (6) limited generalizability to less frequent users or people with unstable MH presentations
Fleury et al. (2019) [42]	Canada	To evaluate the use of and satisfaction with EDs and other MH services among patients with MH disorders, as well as specific characteristics of patient satisfaction and dissatisfaction	Patients with MH disorders	Descriptive convergent mixed methods design	Qualitative interviews with a descriptive questionnaire with quantitative and qualitative components	*n* = 328 (Response rate 88%)	Patients were satisfied with staff attitudes, and sources of dissatisfaction were from information received in EDs regarding community resources and aspects of the physical environment of the ED	(1) Limited generalizability to other EDs settings; (2) older adults underrepresented; (3) differences in experience by MH diagnoses were not collected; (4) no comparator group
Fleury et al. (2019) [44]	Canada	To identify the contributions of predisposing, enabling and needs factors in ED use among patients with mental disorders	Patients with MH disorders	Descriptive cross-sectional mixed methods study	Semi-structured qualitative interview + quantitative questionnaire with sociodemographic, socioeconomic, patient health beliefs, self-assessed health, and ED utilization and satisfaction questions	*n* = 328 (Response rate 88%)	Predisposing factors: being single, of low socioeconomic status, and adequate knowledge of MH services. Enabling factors: having a regular source of care. Needs factors: self-rated importance of problems and having a MH diagnosis of suicidal ideation/attempts, depression, anxiety, and substance use disorders	(1) Limited generalizability to non-similar health systems, EDs of different demographic makeup, and certain diagnostic categories not represented; (2) differences in experience by MH diagnoses were not collected; (3) no comparator group
Wise-Harris et al. (2016) [45]	Canada	To explore the experiences of patients with mental illness and addictions who frequently present to hospital EDs	Patients with mental illness and addictions who frequently present to hospital EDs	Descriptive qualitative—phenomenological study of participants within the treatment arm of a randomized controlled trial (RCT) comparing treatment as usual with a brief case management intervention	Quantitative surveys + qualitative interviews within intervention group	RCT *n* = 166 65% had a mood disorder, 29% a psychotic illness, 42% a concurrent alcohol use disorder and 28% a substance use disorderQualitative interviews *n* = 20 (all within the intervention group)	Participants described their ED visits as being unavoidable and appropriate, despite perceptions of stigmatization and being discharged without treatment	(1) Differences in experience by MH diagnoses were not collected; (2) underrepresentation of individuals who are vulnerably housed and members of ethnic minority groups; (3) limited generalizability to individuals who are non-frequent users of the ED; (4) no comparator group
**Provider Perspectives**								
Betz et al. (2013) [30]	USA	To (1) describe ED provider knowledge, attitudes, and practices related to assessment of suicidal patients, including perceptions of suicide screening; and (2) examine whether these reported factors vary by provider type.	ED nurses, staff/attending physicians, resident physicians	Descriptive cross-sectional study design	Survey collecting quantitative data on participant demographics and knowledge, attitudes, and practices related to the care of suicidal patients	Nurse = 306 Staff/attending physician = 138 Resident physician = 187(Response rate 79%)	Most providers reported deficiencies in (1) knowing how to screen for suicidality; (2) Confidence in skills to assess suicide risk, create a safety plan, provide brief counselling, or provide referrals; (3) MH and administrative support	(1) Subject to self-reporting bias; (2) limited generalizability to non-similar EDs; (3) older adults and members of ethnic minority groups underrepresented; (4) no true comparison group
Chang et al. (2012) [32]	USA	To ask psychiatric providers for their perspectives on the rate-limiting steps in patient care in the ED for adults aged 18 and over who required a psychiatric consultation and to compare them to the patient’s actual length of stay	Fellows/residents, nurse practitioners, social workers, attending staff, and other staff who cared for the patients.	Analytic prospective cohort study comparing provider perspectives with a patient’s actual length of stay	Quantitative analysis and comparison of patient medical records with provider completed log encounters on rate limiting steps in patient encounters	1092 patient encounters Fellow/resident = 521 Nurse practitioner = 30 Social worker = 326 Attending staff = 62 Other = 84	Five rate-limiting steps were identified: (1) limited availability of staff; (2) limited availability of beds after discharge; (3) need for clinical stability; (4) need for additional history; (5) patient’s financial issues	(1) Limited generalizability of findings to non-similar EDs; (2) children and youth, older adults, and members of ethnic minority groups underrepresented; (3) subject to selection bias; (4) differences in experience by MH diagnoses were not collected
Isbell et al. (2019) [35]	USA	To investigate ED providers’ emotional experiences and engagement in their own recent patient encounters, and perceived effects of emotion on patients, in encounters that (1) elicited happiness, (2) elicited anger, and (3) were with a patient with a MH concern	Experienced ED nurses and physicians	Analytic quasi-experimental study design within the framework of a convergent mixed methods study	Mixed-methods survey was administered to collect quantitative and qualitative data on participant experiences. Tasks completed in one of two orders, with all participants describing the mental health encounter last.	Nurse = 44 Physician = 50	Emotions reported in angry and MH encounters were very similar, negative, and associated with low provider engagement compared with positive encounters. Emotions influenced provider’s behaviours and clinical decision-making more in angry and MH encounters.	(1) Subject to self-reporting, social desirability, and recall bias due to method of data collection; (2) differences in experience by MH diagnoses were not collected
Plant and White (2013) [36]	USA	To explore and describe ED nurse’ experiences and feelings caring for patients with mental illness	ED nurses	Descriptive Qualitative—Phenomenological	Qualitative focus groups	Nurse = 10 (reported low response rate) Focus groups = 4	Four themes emerged with an overarching theme of powerlessness: (1) facing the challenge; (2) struggling with the challenge; (3) unmovable barriers; and (4) sinking into hopelessness and seeking resolutions	(1) Subject to selection bias; (2) males underrepresented; (3) small sample size; (4) lack of objective measures of experiences; (5) no comparison group; (6) differences in experience by MH diagnoses were not collected
De Benedictis et al. (2011) [40]	Canada	To examine whether staff perceptions of factors related to the care team and violence on the ward predicted use of seclusion and restraint in psychiatric wards	Nurses, rehabilitation instructors, and nurse’s aides	Analytic cross-sectional study comparing low and high users of seclusion and restraint	Questionnaire collecting quantitative data on socio demographic variables, team climate, perception of aggression, organizational factors, and measures of seclusion and restrain.	Low users of seclusion and restraint = 135 Higher users of seclusion and restraint = 174	Staff perceptions of aggression, aspects of the team climate, and organizational factors were associated with greater use of seclusion and restraint	(1) Number of total participants listed does not equate to the breakdown of individual providers; (2) subject to recall, social desirability, and self-reporting biases; (3) differences in experience by MH diagnoses were not collected
Fleury et al. (2019) [43]	Canada	To explore barriers and facilitators in MH patient management in four Quebec (Canada) EDs that used different operational models	Managers, physicians, ER, and addiction liaison team members	Descriptive qualitative design—case study	Semi-structured interview + questionnaire on patient characteristics and ability to diagnose and treat MH and substance use disorders	*n* = 49 Psychiatrist = 8 Nurse—22 Manager = 14 Emergency physician = 1 Social worker = 2 General practitioner = 2(100% response rate)	Barriers and facilitators affecting management of patients that were identified include (1) health systems; (2) patients; (3) organizations; (4) from professionals themselves	(1) Small sample size; (2) lack of objective measures of experiences; (3) no comparison group
**Both Patient and Provider Perspectives**								
Clarke et al. (2007) [39]	Canada	To determine MH patient and their families’ satisfaction with care received in regional EDs, with particular emphasis on their evaluation of the role of the psychiatric emergency nurse	Patients with MH concerns, their family members, and HCPs	Descriptive qualitative study—case studies	Focus groups	Patients = 27 Family members = 7 Providers = 5 Patients—self reported diagnoses of psychiatric and depressive illnesses, PTSD, personality disorder, and co-occurring mental health and substance use disorders	Key themes identified in the analysis included: (1) waiting in the ED; (2) attitudes of treatment staff; (3) diagnostic overshadowing; (4) nowhere else to go; (5) family needs; (6) A wish list for ideal services.	(1) Limited generalizability to less frequent users of the ED, rural settings, and First Nations peoples; (2) small sample size; (3) lack of objective measures of experiences; (4) no comparison group; (5) selection bias for B negative experiences; (6) lacks ED provider perspectives; (7) differences in experience by MH diagnoses not collected

## Data Availability

Not applicable.

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
