# Peer review of "Patient and Provider Perspectives on Emergency Department Care Experiences among People with Mental Health Concerns"

_healthcare, 2022, doi:10.3390/healthcare10071297_

Round 1

Reviewer 1 Report

This is an interesting study summarizing the care of people with mental health concerns from two perspectives: patients and providers. Although the manuscript is well written and the results are interesting, I have some concerns about the methods.  

Could you please provide more information of the Methods and analysis, how did you get to the study results? where was the structured checklist from? Which scientific analysis was used to identify themes and sub-themes? Please describe the process of the qualitative assessment in more detail so that the reader exactly understands the research done.

In addition, concrete research questions and hypotheses are missing, please include in the last paragraph of the Introduction

In addition there are some minor points:

·        - Please provide prevalence rates of MH disorders in North America.

·       -   the abbreviation PMHC sounds akward, I would suggest people with MH concerns

·      -    it is often one space too many, e.g. line 35, 40, 64, 66, 120, 480, 439

·       -   the authors switch between HCPs (line 63) and HCP (line 78), please be consistent

·         - sometimes there is an Oxford comma (e.g. line 82, 186) and sometimes not

·        -  Be consistent patients with HCP once introduced this abbreviation; Be consistent people with MH concerns (especially in Table 1) as it stands now that those have many names patients, people, respondents, subjects, persons

·         - The mix of studys investating patients vs. providers is somehow confusing in Table1, maybe 2 separate Tables would be better

·  -   What is the difference between attitudes, beliefs and perspectives. Please explain and clarify.

Reviewer 2 Report

Thank you for the opportunity to review this paper. It is a clearly written and very informative manuscript. I think it is a valuable academic contribution.  I have a few minor suggestions that I hope will improve the manuscript prior to publication.

Intro

People with mental health concerns (PMHC) – the whole manuscript is very heavy on initialisations but this one and (TIC) in the discussion felt like they decreased my readability and clarity. Please review and see where words are best placed to retain your sentence clarity.

Suicide gestures? I am unfamiliar with this term and I found it bordering on deficit based. Can you consider removing it or help me understand where it is used in the literature.

Section 2.3

Adjudication by team consensus – who are the team? Consider using initials to confirm.

Two researchers (C.N. and L.W.) conducted iterative reviews of the study results to inductively identify common emerging themes and sub-themes of factors that influence ED care among PMHC.

 I question if this belongs under methodological quality? Perhaps a subheading methodological approach/ analysis – more information is needed on the thematic approach and process too please.

Discussion The discussion seems to move into a summary of the results rather than a presentation of the results in relation to the current literature/health care setting/ policy landscape. Small tweaks and increased referring would quickly resolve this issue.

Very heavy on the limitations, I think you could tone down on a few of them and please include some strengths of your paper too.

Again, this is a great paper. I look forward to reading the revised manuscript.

Round 2

Reviewer 1 Report

The authors responded adequately to the points raised and significantly improved the manuscript.